REGISTERED REPORT PROTOCOL

# Registered report protocol: Factors associated with inter-rater agreement in grant peer review

Jan-Ole Hesselberg[1,2*], Pål Ulleberg[1], Øystein Sørensen[1], Knut Inge Fostervold[1], Sigrid Hegna Ingvaldsen[2], Ida Svege[3]

1 Department of Psychology, University of Oslo, Oslo, Norway, 2 The Foundation Dam, Oslo, Norway, 3 NIFU Nordic Institute for Studies of Innovation, Research and Education, Oslo, Norway

* post@janolehesselberg.no

## Abstract

Grant peer review processes are pivotal in allocating substantial research funding, yet concerns about their reliability persist, primarily due to low inter-rater agreement. This study aims to examine factors associated with agreement among peer reviewers in grant evaluations, leveraging data from 134,991 reviews across four Norwegian research funders. Using a cross-classified linear regression model, we will explore the relationship between inter-rater agreement and multiple factors, including reviewer similarity, experience, expertise, research area, application characteristics, review depth, and temporal trends. Our findings are expected to shed light on whether similarity between reviewers (gender, age), their experience, or expertise correlates with higher agreement. Additionally, we investigate whether characteristics of the applications—such as funding amount, research area, or variability in project size—affect agreement levels. By analyzing applications from diverse disciplines and funding schemes, this study aims to provide a comprehensive understanding of the drivers of inter-rater agreement and their implications for grant peer review reliability. The results will inform improvements to peer review processes, enhancing the fairness and validity of funding decisions. All data and analysis scripts will be publicly available, ensuring transparency and reproducibility.

## Introduction

### Background

Enormous funds are distributed through grants that rely on peer review processes to select which projects to fund. According to the latest review on grant peer review more than 95% of academic medical research funding is awarded through peer review processes [1]. In the US, the National Institutes of Health (NIH) invests around USD 50 billion every year in health research alone, about 80% of them through competitive grants [2], and in the EU nearly EUR 100 billion will be distributed through the Horizon Europe initiative [3]. In Horizon 2020, the predecessor of the Horizon Europe initiative, over 20.000 expert reviewers were involved in reviewing applications [4].

**Data availability statement:** The primary data set will be made available together with the final publication under the CC BY-SA 4.0 licence at kaggle.com/datasets/janolehesselberg upon study completion.

**Funding:** The study is part of JOHs PhD. JOH is the Chief Program Officer at the Norwegian Foundation Dam (www.dam.no) and the PhD is funded through his salary at the foundation. The foundation has also contributed data to the study. The funders had no role in study design, data collection and analysis, decision to publish, or preparation of the manuscript.

**Competing interests:** JOH and SHIs roles as the Chief Program Officer and Senior Advisor at one of the funders (Foundation Dam) in the study might be perceived as a competing interest. This does not alter our adherence to PLOS ONE policies on sharing data and materials.

However, there is considerable debate about how well peer review predicts the future impact of a study. In the most recent review of the literature, authors conclude that "There is (..) fairly clear evidence that peer review is, at best, a weak predictor of future research performance" [1]. The authors cite several possible reasons why this is the case and highlight the problem of low agreement between reviewers scoring the same application as one of the main reasons.

Disagreement is not necessarily a problem. No reviewer can be expected to have the complete overview of a field. Different peers will bring different expertise and experience to the review. This could create a type of disagreement, which to a certain extent is desired. Furthermore, differing opinions on legitimate academic debates could also result in a disagreement which is not necessarily undesirable [5,6]. If this is taken into account, a very high level of agreement could indicate that the reviewers are not diverse enough, consequently leading to important aspects being missed in the review process. Clearly, agreement is in itself not sufficient to secure the quality of grant peer review. But there are degrees of agreement and even in the utopian case of all disagreement being of the desirable type, there are limits to how much disagreement can be tolerated in a decision-making process overall. Low levels of agreement translate directly into low reliability in the overall process and without adequate levels of reliability, the validity of peer review is threatened [7]. To use a metaphor, if you have a thermometer and only can take two measurements of your body temperature you might trust it if it varies by 0.3°C in but if it varies by 3°C you will certainly not trust it, regardless of how good the reason for the variability is.

Several studies have shown that grant reviews often have poor reliability, and almost never reach what is considered acceptable or "good" reliability, commonly defined as an intraclass correlation of 0.75 [7,8]. In the case of grant peer review, an intraclass correlation of 0.75 can be interpreted to mean that 75% of the variance in the ratings given by peer reviewers is attributable to the applications. Hence 25% are attributable to other factors than the applications, like attributes of the reviewers (for example their scoring styles or their school of thought), or other situational attributes (for example the order the applications are reviewed in or if and how panel discussions are run).

One of the first studies that assessed the agreement of grant peer review found single rater intraclass correlations between 0.17 and 0.37 in the analyses of reviews of applications to the National Science Foundation (NSF) in the United States [9]. Later studies have produced similar results. A study of 23.414 ratings in the Austrian Science Fund found an intraclass correlation of 0.26 [10] and a study based on ratings at a Norwegian funder found an intraclass correlation of 0.29, even after implementing an intervention to increase the agreement of their review processes. Using the Spearman-Brown formula [11] one can calculate how many reviewers would be needed to pass the threshold for an intraclass correlation of 0.75 for the review process as a whole, and for the mentioned values, this would be between 6 and 15 reviewers.

Low levels of agreement means that other factors than the applications matter for the outcome, and in a survey of 10,023 ratings by the Australian Research Council

the authors conclude that the decision of whether or not an application was granted, was based substantially on chance [8]. In the same paper, the authors highlight the gravity of the problem and claim that the "most basic, broadly supported, and damning criticism" of grant peer review "is its failure to achieve acceptable levels of agreement among independent assessors". This problem of low agreement in grant peer review seems to generalize across different academic disciplines [8,10].

Several studies have explored factors associated with peer reviewers ratings in grant peer review [1,8] and several studies have addressed single factors that are associated with agreement [1,12,13]. However, despite the problem of low agreement being both well documented and widespread, there is a lack of studies with the aim of identifying how multiple factors contribute to low agreement. To our knowledge only the above-mentioned study that used data from 23,414 ratings at the Austrian Science Fund [10] has done this. The results showed that the research area of the proposed project was associated with agreement and that the other factors included in the study, such as the application amount, the gender of the applicant, the gender of the reviewer and the age of the applicant, were not significantly associated with agreement.

There will always be significant differences between funders that might affect agreement. Differences in review criterias, review type, reviewer experience, the size of the proposed research projects, the scope of the funding programs, and many other factors might make associations that are found at one funder disappear at other research funders. There is a need for larger studies, including more funders and more explanatory factors to assess the robustness of previous findings and to test other hypothesised associations [1,14]. In this study, we will use data from 134,991 reviews at four different Norwegian research funders. To our knowledge this will be the first study on factors associated with agreement, that use data from multiple funders and that has predetermined research questions, methods and code.

### Research questions

**1. Are similarities between reviewers associated with agreement?** One possible hypothesis is that reviewers have a tendency to give higher ratings to applicants that are similar to themselves, and that this should result in more agreement in groups of reviewers that are similar. This has been called the "matching hypothesis" [8] and it has some support in psychological research indicating that we tend to develop a preference for things we are exposed to frequently. The effect is known as the mere-exposure effect or familiarity-breeds-liking [15].

Research addressing this question in peer review shows mixed conclusions. Researchers analyzing reviews from the Frontiers series of journals found "a substantial same-gender preference (homophily), and that the mechanisms of this homophily are gender-dependent" [16], while a study using data sets from the using Australian Research Council found no support for the hypothesis in terms of gender, academic title or age [8,12,17].

Based on the previous findings, we expect that reviewers that are similar in their characteristics, will agree more.

**2. Is reviewer experience of reviewing positively associated with agreement?** The experience of reviewers varies considerably. Some will have taken part in multiple calls and might have reviewed several hundred applications, while others have no experience. Previous research has shown that reviewers that rate many applications are both stricter and more reliable than those who rate few [8,12,18].

In an analysis of over 50 000 applications reviewed in two funding programmes in the European Union from 2014 to 2018, Seeber and colleagues concluded that "reviewing experience on previous calls of a specific scheme significantly reduces disagreement" [19].

Based on these findings we expect the reviewer's experience to be positively associated with levels of agreement.

**3. Is the research area associated with agreement?** Several studies have found that reviewer scores vary considerably between the applications´ funding program and research topic [20]. A study done by Mutz and colleagues of 23.414 ratings in the Austrian Science Fund found that applications in programs related to the humanities have the highest level

of agreement, while biosciences have the lowest levels [10]. The researchers speculate that this "may be due to the lack of uniform evaluation standards or to greatly varying quality of applications".

Based on these results we believe applications within the humanities and social sciences will have higher levels of agreement than those within research areas associated with technical, natural, medical or bio sciences.

**4. Is the application amount associated with agreement?** Different funding schemes allow for a wide variety of application amounts. It is likely that there are systematic differences between applications with different application amounts. Some of these differences are likely to affect levels of agreement, in various ways and directions.

We suspect that some are likely to increase agreement as the application amounts increase. For example, reviewers of applications with large amounts are probably more experienced both as reviewers and researchers, devote more time to the review and are more likely to have been part of panel decisions that give them the chance to calibrate scoring styles against other reviewers.

On the other hand, there are reasons to believe that some factors associated with applications with smaller application amounts also contribute to higher agreement. For example, applications in programs with smaller application amounts are likely to be shorter. Less information might make it more likely that the reviewers focus on the same information when assessing the application. This again, might leave less room for disagreement. Shorter applications might also lower the bar for applicants, resulting in higher variability in application quality. Increased variability in quality should make it easier to distinguish between good and poor applications, and thereby increase agreement.

This question of application amounts being associated with agreement was included in the abovementioned study by Mutz and colleagues. The results showed a significant, positive association with agreement - higher application amounts, was associated with higher agreement - but the authors found the practical importance of the results to be negligible [10]. We believe these findings will be replicated in our study.

**5. Is the variability of project size within funding programs associated with agreement?** Some programs will accept a wide range of projects in terms of the application amount, while other programs will only accept a narrow range of application amounts. The similarities between the projects in programs with a narrow range of accepted amounts will likely be greater than in the programs with greater variability. We believe less similarities between the projects will make it easier to distinguish the applications from each other in terms of quality, resulting in higher agreement.

**6. Is the review type associated with agreement?** There are differences in the responsibilities of reviewers assessing the same application. Some funders use primary and secondary reviewers, where the primary reviewers typically are expected to do an in-depth review. Some funders also use external, hand-picked experts that will get applications within their field of expertise. These specialists are usually also expected to do in-depth reviews. We believe these in-depth reviews will increase agreement, compared to the reviews that are less in-depth.

**7. Is the reviewers level of specialist knowledge associated with agreement?** As mentioned above, some reviewers are hand-picked to review applications within their specific field of expertise. Other reviewers are part of panels that review many applications within a single call for proposals. Reviewers in the latter setting are likely to review some applications that are outside their field of expertise. We expect the hand-picked expert reviewers to agree more, than reviewers that are part of panels.

**8. Is there a trend over time towards higher agreement in grant peer review?** It might be expected that the agreement in grant peer review has increased over the years. As the introduction shows, the problem of low agreement has been in the spotlight for a long time. Most research funders are probably aware of the issue and as funders gradually become better at designing review processes, instructions and reviewer training, increased agreement is a reasonable assumption. However, other factors - like changed criterias, turnover in reviewers, budget cuts and so on - might pull in the other direction and contribute to a decrease in agreement.

Mutz and colleagues [10] found no systematic trend in changes in agreement over the years from 1999 and 2009, and we expect to replicate this finding in our study.

## Methods

### Data

**Data collection procedures.** Data was collected directly from four different Norwegian funders:

1. The Foundation Dam (Dam). Data retrieved from the grant management system Insights Grants by JOH June 4th 2024.

2. The Kavli Trust (KT). Data retrieved from the application system SurveyMonkey Apply by JOH June 4th 2024.

3. The Norwegian Cancer Society (NCS). Data was retrieved by staff at the NCS from their grant management system Insights Grants and emailed to JOH January 27th 2024.

4. The Norwegian Research Council (NRC). Data was retrieved by staff at the NRC from their grant management system and emailed to JOH January 27th 2024.

One of the authors (JOH) has had access to the complete data set. The data set does not contain information that could be used to identify individual participants directly, but contains information that in combination with other public information might make this possible. The use of the data has been approved by the Norwegian Agency for Shared Services in Education and Research (application number: 764354).

The data has been partially observed to retrieve basic information on the contents, but key variables have not been observed and no statistical analysis has been performed prior to writing this protocol. This corresponds to a level 2 bias control, according to the Peer Community Inn Registered Report standards [21].

**Sample.** The primary data set, before any exclusion criteria are implemented, consists of 134,991 reviews of 28,573 applications, done by a total of 6,014 reviewers, between 2013 and 2024. All applications have been reviewed on a scale from 1 (poor) to 7 (excellent). Table 1 shows the distribution of data between the four funders.

The codebook describing the primary data set in detail can be found in the supplementary material (see S1 Table).

**Data structure.** The data has a cross-classified structure with the following important properties at different levels (see Fig 1):

• **Level 4 - Funder.** Four different funders with different programs.

• **Level 3 - Programs.** The funding programs are exclusive to funders.

• **Level 2a - Applications** are exclusive to programs. They are all reviewed by at least two reviewers but might be reviewed by anywhere from two to nine reviewers.

• **Level 1 - Agreement score.** The agreement between reviewers assessing the same application.

• **Level 2b - Reviewers** are exclusive to funders but might operate under more than one program. In addition, they might review more than one application, sometimes more than a hundred.

**Table 1. Contents of the primary data set.**

| Funder | Years, range | # of programs | # of applications | # of reviews | # of reviewers |
|---|---|---|---|---|---|
| Dam | 2020–2024 | 1 | 1,735 | 8,680 | 50 |
| KT | 2017–2024 | 1 | 297 | 1,339 | 12 |
| NCS | 2017–2023 | 1 | 1,105 | 6,659 | 115 |
| NRC | 2013–2023 | 93 | 25,436 | 118,313 | 5,837 |
| **Total** | **2013–2024** | **96** | **28,573** | **134,991** | **6,014** |

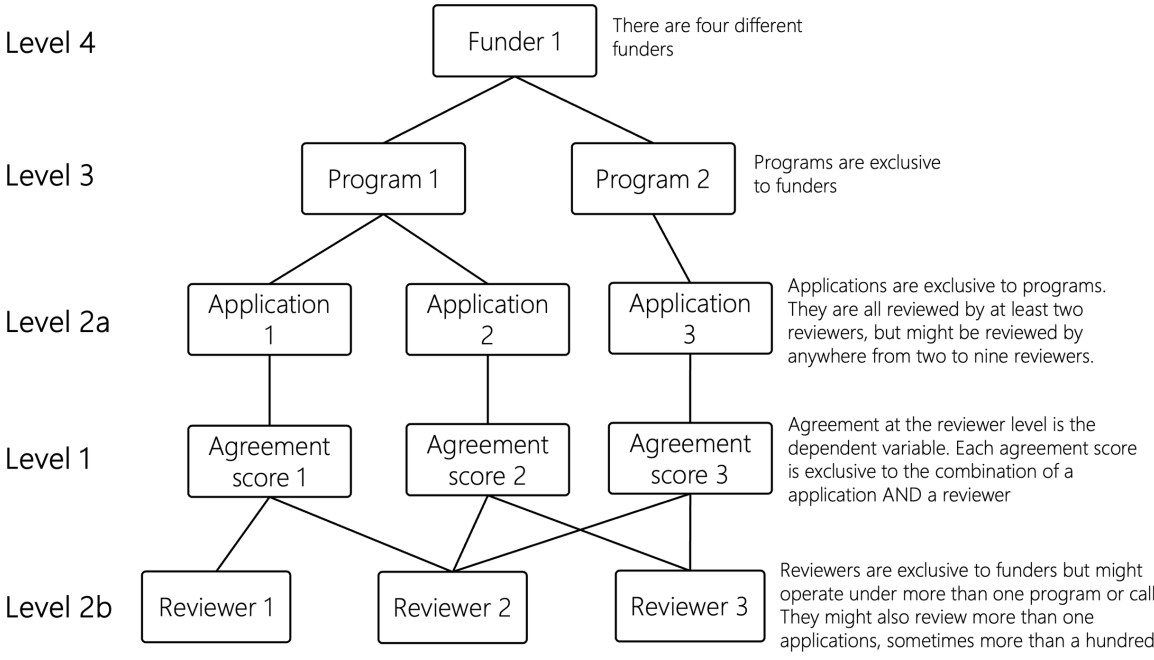

**Fig 1. Data structure.**

## Variables

**Response variable.** The response variable is agreement between reviewers of the same review type assessing the same application. Hence, only applications with two or more independent review scores of a particular review type (external expert, panel member or depth reviewer) will be included in the analyses.

Agreement will be assessed by using the Average Deviation (AD) index. The AD index shows how much a reviewer's assessment of an application deviates from the mean of the individual scores of the application. When the AD index is zero, the raters are in complete agreement.

The following formula will be used:

$$Disagreement_{ij} = \left| Score_{ij} - \frac{1}{n_i} \sum_{j=1}^{n_i} Score_{ij} \right|$$

In the formula, *i* represents the application and *j* the reviewer. Hence $Score_{ij}$ represents the reviewer's score of an application. *n* stands for the number of reviewers (*j*) assessing a specific application of a specific review type (*i*)

In this study, the reviewers score on an application ($Score_{ij}$) will either be a overall score set by the reviewer or, where this is not available, an aggregate score calculated on the basis of the reviewers criteria scores. All applications and review criteria are scored on a scale from 1 (poor) to 7 (excellent).

We chose to use the AD index as our primary indicator of inter-rater agreement instead of alternatives such as the intraclass correlation (ICC) or the within-group agreement index ($r_{wg}$) because AD index does not require a specified null distribution and quantifies disagreement in the original scale's units, making it more straightforward to interpret. Furthermore, it has been used in other important studies of agreement in grant peer review [22,23] and simulation research has demonstrated that the AD index performs favorably when compared with other inter-rater agreement indices [24–26].

**Explanatory variables.** We will produce nine explanatory variables that include different attributes of the funder, programs, applicant, application, reviewer and the review. For an overview of the explanatory variables, their coding and the predicted associations see Table 2. The numbering of the explanatory variables correspond with the numbering of the research questions. The steps taken to produce the explanatory variables are described in detail in the attached R script (see S2 Script).

**1a. Share of reviewers of same gender:** The share of reviewers of the same gender will be calculated for each review of a particular review type (external expert, panel member or depth reviewer) per application. For example: If two out of three reviewers are male, the value returned will be 0.67. The same value will be returned if two out of three are female.

**1b. Standard deviation of reviewer age per application:** The standard deviation of the age of reviewers of a particular review type (external expert, panel member or depth reviewer) per application.

**2. Share of young reviewers:** As we do not know how many reviews the reviewers have done before the first review in the dataset, we do not have a reliable and direct way of assessing the reviewers level of reviewing experience. As a proxy for the variability in reviewer experience within the review of an application, we will calculate the share of reviewers that are among the 20 percent youngest reviewers in the data set, at the time of the review. For example: If two out of three reviewers are among the 20 percent youngest, the value returned will be 0.67.

**3. Research area:** The research area of each application. Seven categories are used: 1 Humanities, 2 Social sciences, 3 Technology, 4 Medicine and health, 5 Mathematics and natural sciences, 6 Agriculture and fish, 7 Other. Each

**Table 2. Explanatory variables, coding and predicted associations.**

| Variable | Coding | Predicted association |
|---|---|---|
| 1a. **Reviewer similarity, gender** | Share of reviewers of same gender, 0.5–1.0 | **+ Positive association** Increased similarity = Increased agreement |
| 1b. **Reviewer similarity, age** | SD of reviewer age per application | **+ Positive association** Increased similarity = Increased agreement |
| 2. **Share of young reviewers** | Share of reviewers that are among the 20 percent youngest reviewers in the data set | **- Negative association** Higher share of young reviewers = Decreased agreement |
| 3. **Research area** | 3a Humanities TRUE/FALSE 3b Social sciences TRUE/FALSE 3c Technology TRUE/FALSE 3d Med and health TRUE/FALSE 3e Math and natural TRUE/FALSE 3f Agriculture TRUE/FALSE 3g Other TRUE/FALSE | **Association** There will be significant differences between different research areas. Humanities and Social sciences will have higher levels of agreement than Technology, Math and natural science, and Medicine and health |
| 4. **Application amount** | Amounts in million NOK | **+ Positive association** There will be a statistically significant association between application amount and agreement, but it will probably not be of practical importance |
| 5. **Variability in application amount of program** | Coefficient of variation - SD of applied amount in program divided by average of applied amount in program | **+ Positive association** More variability = Increased agreement |
| 6. **Review depth** | TRUE = depth review FALSE = panel member (less depth) | **+ Positive association** There will be higher agreement for in-depth reviews than for panel member reviews |
| 7. **Reviewer specialist knowledge** | TRUE = specialist FALSE = non-specialist | **+ Positive association** There will be higher agreement for specialist reviews than for non-specialist reviews |
| 8. **Year of submittal** | The year the application was submitted (2013–2024) | **0 No association** There will not be a significant trend towards more or less agreement over the years |

of the categories will be coded as a logical variable. It is possible for an application to fall under more than one category. All applications from the Cancer society, the Kavli Trust and the Foundation Dam, fall under the "Medicine and health" category.

**4. Application amount:** The requested grant sum of the application. No calculation is required for this variable. The amounts are in million Norwegian kroner (NOK).

**5. Coefficient of variation in application amount of program:** To capture the variability of application amounts per program, we will calculate the coefficient of variation (CV) by dividing the standard deviation of application amount (in million NOK) per program by the mean application amount per program.

**6. Review depth:** There are three review types in the data set: External expert, Panel member or Depth reviewer. External experts and Depth reviewers are both performing more in-depth reviews than the panel members. This variable will be coded TRUE if the review type is External expert or Depth review, and FALSE if the review type is Panel member.

**7. Reviewer specialist knowledge:** External reviewers are usually experts within the topic of the application. This variable will be coded TRUE if the review type is External expert and FALSE if the review type is Panel member or Depth reviewer.

**8. Year of submittal:** To assess whether or not there is a trend towards more or less agreement over time in the data set, we will use the year the application was submitted (from 2013 to 2024).

## Analysis plan

**Descriptives statistics.** We will provide a table describing the number of entries removed in the different steps of data exclusion, and a table with a row for each funder and the following descriptive statistics per funder:

• Number of programs

• Number of applications

• Number of reviews

• Average number of reviews per application

• Number of reviewers

• Average score and SD of average scores

• Average deviation and SD of average deviation

**Statistical models.** To test the hypotheses, we will use a cross-classified linear regression model. Similar approaches have been used by Jayasinghe and colleagues [12] and Seeber and colleagues [23]. This approach accounts for the fact that the disagreement scores (the response variable) on the one hand are nested in applications within programs within funders, and on the other hand in reviewers across applications and programs. The method also considers that the relationship between the explanatory variables and the response variable (disagreement) may be influenced by unobserved variables impacting both. Since reviewers can evaluate multiple applications, the structure is not hierarchical but cross-classified. If we experience problems with convergence, we will systematically reduce the number of random effects. The distribution of scores for each explanatory variable will be examined prior to inclusion in the analyses. Variables that deviate considerably from a normal distribution will be considered for transformation to achieve a more normal distribution before inclusion.

The analyses will be performed using R 3.6.0+, RStudio version 2024.04.2+764 [27] and the lme4 package version 1.1–35.5 [28].

**Inference criteria.** We will use the standard $p < .05$ criteria for determining if the linear regression coefficients for the hypothesized factors have an effect on the agreement between reviewers. For interpreting effect sizes we will use the estimated regression coefficients and their associated 95% confidence intervals.

We expect that some statistically significant results will have little practical significance, but we will not predetermine criteria for inferring the practical significance for the hypothesized factors. The main reason is that the practical significance will vary with who the end user is. A weak association might be of importance to the funder, and have even greater implications for the society as a whole, but have a negligible impact on the review of individual applications. Additionally, we are not aware of reliable ways to determine the level for practical significance of differences in agreement, given these different end users perspectives. For this reason we will leave the reflections on practical significance to the discussion of the results.

**Data exclusion.** The following entries will be removed in the following order before analysis:

1. **Unlikely reviewer ages.** Entries of reviewer ages below 21 and above 100 will be removed.

2. **Duplicate entries**. It is not expected that there will be many, if any, duplicate entries.

3. **Missing review scores.** Rows where review scores are missing will be removed.

4. **Impossible review scores**. Rows containing review scores below the minimum score of 1 and above the maximum score of 7 will be removed.

5. **Missing application amounts.** Rows with missing application amounts will be removed.

6. **Unlikely application amounts**. Rows containing application amounts that are more than ten times higher than the average amount of the program will be removed.

7. All rows with **less than two reviews** of a particular review type per application. At least two reviewers are needed to calculate the response variable. It is expected that a significant share of the entries in the data set will be excluded based on this criteria.

8. All rows related to **programs with less than ten applications.** Some programs will have very few applications. These programs are likely to have some attributes that make the applications in them especially hard to compare with other applications. In addition, as one of the explanatory variables is variability in application amount of programs, we will need some applications to calculate a reliable value. Therefore, we will only include programs with 20 or more applications. We do not expect many programs to have less than 20 applications.

**Missing data.** We will exclude all rows with missing values to ensure the integrity and interpretability of the results. The cross-classified linear regression model benefits from complete data for all variables to accurately estimate parameters and account for the nested and cross-classified structure of the data. Excluding rows with missing values helps maintain consistency in the dataset and prevents potential confounding due to incomplete information. This approach is particularly important given the complexity of our model, which seeks to disentangle multiple levels of relationships and account for unobserved variables.

**Sensitivity analyses.** We will explore the effect of removing missing data by doing regression analyses of each explanatory variable separately, and comparing the results to the main analysis.

**Exploratory analysis.** We will explore possible differences between the funders by performing and ANOVA, and visualize the differences between funders using a box plot. Additionally we will calculate the intraclass correlations (ICC (11)) per funder for the two most common review types (panel member and depth reviewer) combined, and use the average number of reviews per funder to calculate the average measure intraclass correlation (ICC (1,k)).

**Codebook and script.** The codebook describing the contents of the primary data set is included in the supplementary materials (see S1 Table). Two R scripts are used in this study, a preprocessing script (see S1 Script) and the analysis script (see S2 Script). The preprocessing script merges the data from the four funders, produces the explanatory variables based on personal data, removes personal data and exports the primary data set for analysis. The analysis script produces the remaining explanatory variables and runs the analysis described above.

## Supporting information

**S1 Table. Codebook for primary data set.** Codebook describing the data in the primary data set used as the starting point for the analysis script (see S2 Script).
(XLSX)

**S1 Script. Preprocessing script.** This script has been used for producing the primary data set used as the basis for the analysis script (see S2 Script). This includes merging the data sets from the four funders and producing the explanatory variables based on variables that could be used to identify reviewers or applications (gender and birth year). Additionally, all application IDs and reviewer IDs are switched with anonymized IDs.
(R)

**S2 Script. Analysis script.** This script describes the data exclusion, the production of the response variable, the remaining explanatory variables, removing of missing data, main analysis, sensitivity analyses and the production of the final codebook.
(RMD)

## Acknowledgments

We wish to thank Secretary General Hans Christian Lillehagen at the Foundation Dam for providing financial and in-kind support. In addition, this project would not be possible without the support from Kavli Trust, the Norwegian Cancer Society and the Norwegian Research Council, and their willingness to to share their data and answering many questions regarding the different variables.

## Author contributions

**Conceptualization:** Jan-Ole Hesselberg, Pål Ulleberg, Øystein Sørensen, Knut Inge Fostervold, Sigrid Hegna Ingvaldsen, Ida Svege.

**Data curation:** Jan-Ole Hesselberg, Sigrid Hegna Ingvaldsen, Ida Svege.

**Formal analysis:** Jan-Ole Hesselberg, Pål Ulleberg, Sigrid Hegna Ingvaldsen, Ida Svege.

**Funding acquisition:** Jan-Ole Hesselberg.

**Investigation:** Jan-Ole Hesselberg, Øystein Sørensen, Ida Svege.

**Methodology:** Jan-Ole Hesselberg, Pål Ulleberg, Øystein Sørensen, Knut Inge Fostervold, Sigrid Hegna Ingvaldsen, Ida Svege.

**Project administration:** Jan-Ole Hesselberg, Ida Svege.

**Supervision:** Pål Ulleberg, Øystein Sørensen, Knut Inge Fostervold, Ida Svege.

**Writing – original draft:** Jan-Ole Hesselberg, Pål Ulleberg, Øystein Sørensen, Knut Inge Fostervold, Sigrid Hegna Ingvaldsen, Ida Svege.

**Writing – review & editing:** Jan-Ole Hesselberg, Pål Ulleberg, Øystein Sørensen, Knut Inge Fostervold, Sigrid Hegna Ingvaldsen, Ida Svege.

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
