## [Decision Letter · Decision Letter 0]

14 Mar 2025

PONE-D-25-01557Registered report protocol: Factors associated with inter-rater agreement in grant peer reviewPLOS ONE

Dear Dr. Hesselberg,

Thank you for submitting your manuscript to PLOS ONE. After careful consideration, we feel that it has merit but does not fully meet PLOS ONE’s publication criteria as it currently stands. Therefore, we invite you to submit a revised version of the manuscript that addresses the points raised during the review process.

We look forward to receiving your revised manuscript.

Kind regards,

Ulf Sandström

Academic Editor

PLOS ONE

Journal Requirements:

3. Thank you for stating the following financial disclosure: [The study is part of JOHs PhD. JOH is the Chief Program Officer at the Norwegian Foundation Dam (www.dam.no) and the PhD is funded through his salary at the foundation. The foundation has also contributed data to the study]. 

4. Thank you for stating the following in the Competing Interests section: [JOH and SHIs roles as the Chief Program Officer and Senior Advisor at one of the funders (Foundation Dam) in the study might be perceived as a competing interest.].

We note that one or more of the authors have an affiliation to the commercial funders of this research study. [Foundation Dam].

5. We note that you have indicated that there are restrictions to data sharing for this study. For studies involving human research participant data or other sensitive data, we encourage authors to share de-identified or anonymized data. However, when data cannot be publicly shared for ethical reasons, we allow authors to make their data sets available upon request. For information on unacceptable data access restrictions, please see http://journals.plos.org/plosone/s/data-availability#loc-unacceptable-data-access-restrictions.

6. We are unable to open your Supporting Information file [S1 Script.R and S2 Script.Rmd]. Please kindly revise as necessary and re-upload.

Reviewers' comments:

Reviewer's Responses to Questions

**Comments to the Author**

1. Does the manuscript provide a valid rationale for the proposed study, with clearly identified and justified research questions?

Reviewer #1: Partly

Reviewer #2: Yes

2. Is the protocol technically sound and planned in a manner that will lead to a meaningful outcome and allow testing the stated hypotheses?

Reviewer #1: Partly

Reviewer #2: Yes

3. Is the methodology feasible and described in sufficient detail to allow the work to be replicable?

Reviewer #1: Yes

Reviewer #2: Yes

4. Have the authors described where all data underlying the findings will be made available when the study is complete?

Reviewer #1: Yes

Reviewer #2: Yes

5. Is the manuscript presented in an intelligible fashion and written in standard English?

Reviewer #1: Yes

Reviewer #2: Yes

6. Review Comments to the Author

You may also provide optional suggestions and comments to authors that they might find helpful in planning their study.

Reviewer #1: See attachment, See attachment, See attachment, See attachment, See attachment, See attachment, See attachment, See attachment

Reviewer #2: The report protocol presents a very interesting research plan.

My only observations at this stage regards the following.

1. novelty statements:

“there is a lack of studies with the aim of identifying how multiple factors contribute to low agreement. To our knowledge only the above-mentioned study that used data from 23,414 ratings at the Austrian Science Fund (10) has done this”

and

“To our knowledge this will be the first study on factors associated with agreement, that uses data from multiple funders”

The cited paper from Seeber et al. (2021):

- do consider multiple factors (possibly some more are relevant for this study)

- it combines data from four programs from two funders: 3 from MSCA and one from COST

I agree instead it is the first study with “predetermined research questions, methods and code.”

2. One additional comment regard the dependent variable: perhaps you could consider the gap in score between couples of reviewers. This would provide a more robust test of similarity hypothesis. Suppose we have 3 reviewers, 2 males and 1 female, one could test if the agreement is lower for the 2 males, than the 2 couples m-f.

3. You could also clarify if the analysis is based on which stage of the evaluation – considering that at least DAM has a two-stage process (Seeber, Svege, Hesselberg 2024).

7. PLOS authors have the option to publish the peer review history of their article (what does this mean? ). If published, this will include your full peer review and any attached files.

**Do you want your identity to be public for this peer review?** For information about this choice, including consent withdrawal, please see our Privacy Policy .

Reviewer #1: **Yes: ** Steven Wooding

Reviewer #2: No

---

## [Author Response · Author response to Decision Letter 0]

21 Mar 2025

Is the rationale for the proposed study clear and valid?

The rationale is generally clear and well explained.

The authors helpfully raise the idea of helpful and unhelpful disagreement from line 55. However, they then dispense with it on line 63 stating that ‘even in the utopian case of all disagreement being of the desirable type, there are limits to how much disagreement can be tolerated’ - this seems conceptually wrong. If all the disagreement is because of the different perspectives being bought to bear then this is important and not information that should be thrown away, or considered the fault of the system of assessment. Their metaphor is also problematic - if the only thermometer available varies by 3°C and you need to know your body temperature, then you will use it, but you will have to take many measurements to find the true temperature. I think this section needs a little revision to take into account that there are two forms of disagreement (at least)

1. Reviewers working from the same premise and experience and coming to different conclusions (unhelpful)

2. Reviewers with different experience/expertise who would be expected to see a different set of issues (helpful and must be managed)

The current project only examines and quantifies these two types of disagreement aggregated together - which is still a valuable contribution (it might be possible to separate these forms of disagreement using the dataset the authors have - see suggestion in next section)

Answer from authors (JOH):

Thank you for this valuable feedback. We agree that “helpful” disagreement, stemming from legitimate differences in perspective or expertise, should not necessarily be minimized. At the same time, high levels of disagreement—whatever the cause—will diminish confidence in final funding decisions and will make the application process unreliable.

We think the apparent paradox might be solved by stressing the difference between disagreement in single cases and in the decision-making process overall and have tried to make this clearer (also in the metaphor).

Unfortunately, our dataset does not include enough reviewer-level background information to separate “helpful” from “unhelpful” disagreement in a rigorous manner, and in practice, it is often difficult to draw that line. We will revise the text to clarify that our measure of disagreement inevitably encompasses both desirable and undesirable forms, and we will note the importance of future research aimed at distinguishing these two types more precisely.

Is the protocol technically sound?

The protocol is largely technically sound, I have two main concerns:

1. The level of analysis. My impression (and apologies if I’ve missed something) is that the response/dependent variable (“Disagreement”) is being calculated at the review level (hence the ij subscript); but all the explanatory variables are being calculated at the application level (ie across the pool of reviewers providing reviews for that application). I think the response variable and explanatory variables have to be calculated for the same entities. If “Disagreement” is being calculated at the application level by aggregation this needs to be explained.

Answer from authors (JOH):

Thank you for pointing this out. We agree it is crucial to clarify precisely how the outcome (disagreement) and the explanatory variables align in our analysis.

We do not believe that the outcome (response variable) and all predictors must be computed at the exact same level. In multilevel or cross-classified models, it’s valid to have an outcome measured at one level (e.g., the review level) while incorporating explanatory variables that reflect higher-level or lower-level characteristics (e.g., application-, program-, or reviewer-level factors).

What matters is that each observation in the dataset has a properly aligned value for the response variable and for each explanatory variable.

If set up correctly, the modeling framework (e.g., cross-classified random effects) accounts for the fact that reviews are simultaneously nested within both reviewers and applications.

2. Multiple hypothesis testing. The authors suggest they will use p=<0.05 significance testing - but they are conducting multiple tests (at least 16, one for each explanatory variable and presumably one for each of the seven “Research Area”; and will they test the effect of time in each area?). I think this means they need to correct for testing multiple hypotheses - as with 16 tests and a p=<0.05 stringency they are quite likely to reject a null hypothesis simply by chance. I think there are approaches available to correct for multiple hypothesis testing in frequentist approaches (although I’m am not very familiar with them) - an alternative approach would be to move to a Bayesian approach to analysis.

Answer from authors:

Thank you for pointing this out. We agree that correcting for the multiple comparisons is important and suggest we use Benjamini–Hochberg and adjust the analysis script accordingly add the following to the protocol:

“We will use the Benjamini–Hochberg (BH) approach to correct for multiple comparisons because it controls the false discovery rate rather than the family-wise error rate—providing a balance between discovering genuine associations and avoiding false positives. By contrast, Bonferroni corrections often become overly conservative in settings with many tests, increasing the risk of overlooking real effects (Benjamini & Hochberg, 1995).”

Minor suggestions

1. Grant size is likely to be hugely variable over orders of magnitude, so it might make sense to transform this variable before statistical analysis - eg by using log(grant size) - this would also be true for the ‘variability’ in grant size.

Answer from authors:

Thank you for pointing this out. Yes, we suspect that the data regarding application amount will be right skewed, and that log-transformation is a good solution if this is the case. We suggest we add the following reservation to the methods section: “The distribution of scores for each variable will be examined prior to inclusion in the analyses. Variables that deviate considerably from a normal distribution will be considered for transformation to achieve a more normal distribution before inclusion."

2. Is Research Area being treated as categorical or can one grant be from multiple areas - table at line 374 suggests there might be multiple TRUE Research areas.

Answer from authors (JOH):

Thank you for highlighting this point. Indeed, some applications in our dataset span multiple research areas—for instance, an interdisciplinary grant that combines social sciences and technology. Consequently, we code each area as a separate binary (TRUE/FALSE) variable rather than forcing each application into a single category. This means an application can have multiple “TRUE” values if it legitimately covers more than one area. We will clarify this in the Methods section to avoid confusion.

We have added sub-numbering to the variables in table 2. Each of the variables under 3 Research area are numbered 3a to 3g. This is also done in the analysis script.

Possible additions

1. Different types of disagreement - if information was available on the field of expertise of the reviewers (which is may well not be) then it might be possible to see if reviewers with similar expertise were more likely to agree when reviewing the same application than reviewers with differing expertise. I don’t think this is quite the same as looking to see if applications that have more reviewers of the same type have more similar review scores.

Answer from authors:

Thank you. We agree that this is an important distinction and an interesting research question. However, we do not have the information needed in our data set to the suggested analyses. We will however point out that this is a topic for further research in the discussion.

Will it effectively achieve its aims, and test the stated hypotheses?

If my two concerns about the technical soundness of the protocol are addressed it will test the stated hypotheses.

Is the methodology feasible and detailed enough to make the work replicable?

Yes

General comments

In the introduction the authors use the term ‘reliability’ - but it is not clear what is being warranted by a reliable score - I think the terminology of ‘agreement between reviewers’ which they adopt in much of the rest of the article is clearer. In the next section they do use ‘agreement’ it should probably be changed in the initial section.

Answer from authors:

Thank you for raising this terminology issue. We agree that “inter-rater agreement” may be clearer and less ambiguous than “reliability” in this context. We have made revisions in the introduction to use “agreement” (and explicitly link it to the broader concept of reliability where appropriate) to maintain consistency throughout the manuscript and avoid confusion.

Line 139 - ‘Is reviewer experience of reviewing positively associated with agreement?’ Is clearer

Answer from authors:

Thank you. This has been adjusted

Line 207 - ‘Is the reviewers level of specialist knowledge associated with agreement?’ Is clearer

Answer from authors:

Thank you. This has been corrected throughout the text

Line 302 - I think ‘overall’ score might be better than ‘total’ score, as I don’t think things are being added together/totalled to provide the ‘total’ score.

Answer from authors:

Thank you. This has been adjusted

Figure 1 - there are two 'Level 2’s in the diagram

Answer from authors:

Thank you for pointing this out. We have changed this to Level 2a (application) and Level 2b (reviewer) in the protocol, the figure and the script

---

## [Editor Report · Decision Letter 1]

27 Mar 2025

Registered report protocol: Factors associated with inter-rater agreement in grant peer review

PONE-D-25-01557R1

Dear Dr. Hesselberg,

We’re pleased to inform you that your manuscript has been judged scientifically suitable for publication and will be formally accepted for publication once it meets all outstanding technical requirements.

Kind regards,

Ulf Sandström

Academic Editor

PLOS ONE

---

## [Editor Report · Acceptance letter]

PONE-D-25-01557R1

PLOS ONE

Dear Dr. Hesselberg,

I'm pleased to inform you that your manuscript has been deemed suitable for publication in PLOS ONE. Congratulations! Your manuscript is now being handed over to our production team.

Kind regards,

on behalf of

Professor Ulf Sandström

Academic Editor

PLOS ONE